# Adiponectin, A-FABP and FGF-19 Levels in Women with Early Diagnosed Gestational Diabetes

**DOI:** 10.3390/jcm11092417

**Published:** 2022-04-25

**Authors:** David Karasek, Ondrej Krystynik, Veronika Kucerova, Dominika Macakova, Lubica Cibickova, Jan Schovanek, Martin Haluzik

**Affiliations:** 1Third Department of Internal Medicine—Nephrology, Rheumatology and Endocrinology, Faculty of Medicine and Dentistry, University Hospital and Palacky University, I. P. Pavlova 6, 77900 Olomouc, Czech Republic; ondrej.krystynik@fnol.cz (O.K.); dominika.macakova@fnol.cz (D.M.); lubica.cibickova@fnol.cz (L.C.); jan.schovanek@fnol.cz (J.S.); 2Department of Clinical Biochemistry, University Hospital, I. P. Pavlova 6, 77900 Olomouc, Czech Republic; veronika.kucerova@fnol.cz; 3Diabetes Centre, Institute for Clinical and Experimental Medicine, 14021 Prague, Czech Republic; halm@ikem.cz

**Keywords:** gestational diabetes, adiponectin, adipocyte fatty acid-binding protein, fibroblast growth factor-19

## Abstract

Background: Adiponectin, adipocyte fatty acid-binding protein (A-FABP), and fibroblast growth factor-19 (FGF-19) belong to proteins involved in glucose metabolism regulation. The aims of the study were to compare the plasma levels of these proteins in women with early diagnosed gestational diabetes mellitus (GDM) to those in healthy controls and to investigate their changes during pregnancy after early intervention. Methods: The study was undertaken as a case-control study. Early GDM diagnosis was based on repeated fasting plasma glucose ≥5.1 and <7.0 mmol/L during the first trimester of pregnancy and exclusion of overt diabetes. Age-matched controls comprised healthy pregnant and non-pregnant women. In addition to adipokines, clinical parameters and measures of glucose control were assessed. Results: Women with GDM (*n* = 23) had significantly lower adiponectin and higher A-FABP levels compared to healthy pregnant (*n* = 29) or non-pregnant (*n* = 25) controls, while no significant differences in FGF-19 between the groups were found. The therapeutic intervention shifted adiponectin and A-FABP levels in GDM women towards concentrations of healthy pregnant controls. Adipokines were associated with visceral adiposity and glucose control. Conclusion: Women with GDM showed altered adipokine production even in the first trimester of pregnancy. Early therapeutic intervention not only improved glucose control but also normalized impaired adipokine production.

## 1. Introduction

According to the International Association of Diabetes and Pregnancy Study Groups, the detection and diagnosis of hyperglycemic disorders in pregnancy involve two phases [1]. The first is performed during an initial prenatal visit (usually in the first trimester) to identify women with overt diabetes not diagnosed before their pregnancy. If the results are not diagnostic for overt diabetes but are abnormal (fasting plasma glucose ≥5.1 mmol/L but <7.0 mmol/L), early gestational diabetes mellitus (GDM) is suspected. Therefore, if overt diabetes is excluded, it is recommended that fasting plasma glucose (FPG) levels in early pregnancy ≥5.1 mmol/L are also classified as GDM [1]. The second phase is a 75 g oral glucose tolerance test (OGTT) performed at 24–28 weeks of gestation in all women who have not been previously diagnosed with overt diabetes or GDM to identify the other women with GDM. Higher FPG levels in the first trimester (lower than those diagnosed with overt diabetes) have been reported to be also associated with an increased risk of adverse pregnancy outcomes [2]. As there is not enough data suggesting that generalized testing is beneficial in the diagnosis and treatment of GDM before the usual period of 24–28 weeks of pregnancy, some guidelines recommend early screening in high-risk patients only [3].

Even though the pathophysiology of GDM is not yet fully understood, it generally involves relatively insufficient insulin secretion with increased peripheral insulin resistance developing during pregnancy [4]. Adipose tissue dysfunction is a well-described cause of increased insulin resistance. Emerging evidence of altered adipokine expression in women with GDM suggests an important involvement of dysfunctional adipose tissue and its impaired endocrine function in the development of hyperglycemia during pregnancy [5]. Adiponectin and adipocyte fatty acid-binding protein (A-FABP) belong to abundantly expressed adipokines closely related to insulin resistance [6,7]. The beneficial effects of adiponectin on glucose metabolism include a reduction in inflammation and oxidative stress and improvement in insulin resistance; a protective effect on pancreatic β-cells; an increase in glucose utilization and fatty acid oxidation in skeletal muscles; a reduction in hepatic glucose production; and an increase in insulin-stimulated glucose uptake by adipocytes [6,8]. The main function of A-FABP is the binding of free fatty acids, but it is also involved in the regulation of inflammatory and metabolic processes [7,9]. Animal model studies suggest that A-FABP is an important player in the regulation of glucose homeostasis. Deletion of the A-FABP gene protects mice from insulin resistance and hyperinsulinemia associated with obesity [10]. Secretion of A-FABP from adipocytes can regulate hepatic glucose production and insulin secretion by pancreatic β-cells [7]. Fibroblast growth factor-19 (FGF-19) is a gut hormone with pleiotropic effects. It is secreted mainly by the small intestine in response to feeding. In addition to its role in bile acid homeostasis, FGF-19 activates an insulin-independent endocrine pathway that regulates hepatic protein and glycogen metabolism, inhibits gluconeogenesis, and promotes glucose uptake in adipocytes [11]. Lower FGF-19 levels have been observed in patients with type 2 diabetes mellitus and obese patients, suggesting that it plays a role in weight loss [12].

Early detection of GDM allows timely intervention to normalize blood glucose levels and prevent adverse pregnancy outcomes. We hypothesized that circulating levels of glucoregulatory adipokines were altered in women with GDM even in the first trimester of pregnancy and that therapeutic intervention might improve their adverse metabolic profiles along with impaired adipokine concentrations. To this end, we compared plasma concentrations of adiponectin, A-FABP, and FGF-19 in women with early-diagnosed GDM with those in healthy pregnant women and healthy non-pregnant controls and studied their changes during pregnancy.

## 2. Materials and Methods

### 2.1. Study Design, Inclusion, and Exclusion Criteria

The study was undertaken as a case-control study in accordance with the principles of the Declaration of Helsinki as revised in 2008. It was reviewed and approved by the Ethics Committee of Medical Faculty and University Hospital Olomouc (approval no. 120/17) and informed consent was obtained from all participants. The diagnosis of early GDM was based on repeated FPG levels ≥5.1 mmol/L and <7.0 mmol/L during the first trimester (8–14 weeks) of pregnancy [1]. The exclusion criteria were having type 1 or type 2 diabetes, secondary or genetic types of diabetes, or a history of GDM. All pregnant women were enrolled in the first trimester and followed for the rest of the pregnancy until delivery. Healthy pregnant women had normal glucose levels throughout their pregnancy, including the OGTT at 24–28 weeks of gestation [1]. Non-pregnant age-matched controls comprised healthy women without a personal history of glucose intolerance or diabetes (including GDM or a history of delivering a high birth weight baby, i.e., ≥4.5 kg). All study participants had normal thyroid function.

Participants were asked about their personal and medical history. Their body mass index (BMI), waist circumference, and systolic and diastolic blood pressure (SBP and DBP) were also measured. The BMI was calculated as body weight/body height^2^ (kg/m^2^). Waist circumference was measured while standing, in the middle between the anterior iliac crest and the lower border of the ribs. Pregnant women (with or without GDM) were examined in the first (8–14 weeks), second (24–28 weeks), and third (34–38 weeks) trimesters. Weight, BMI, waist circumference, SBP, DBP, glucose, glycated hemoglobin A1C (HbA1C), and adipokine levels were checked at each visit. Early therapeutic intervention (nutrition therapy—diet containing 175 g of carbohydrate or 35% of a 2000-calorie intake, a minimum of 71 g of protein, and 28 g of fiber; monounsaturated and polyunsaturated fats were emphasized while saturated fats were limited, physical activity—30 min of exercise (medium intensity walking), 5 days/week, and weight controls) was introduced for women with GDM already during their first visit (in the first trimester). Some of them were also treated with insulin to achieve the glycemic target. The cut-off values for the introduction of insulin therapy were as follows: repeated FPG level >5.3 mmol/L; 1-h postprandial glucose level >7.8 mmol/L, or 2-h postprandial glucose level >6.7 mmol/L [13].

### 2.2. Laboratory Analyses

Venous blood samples were collected in the morning after a 12-h fast. Routine serum biochemical parameters (glucose, HbA1C, and C-peptide) were analyzed on the day of blood collection. Concentrations of adipokines were measured in sample aliquots stored at −80 °C for no longer than 6 months.

Glucose levels were determined using the hexokinase method (GLUC3, Roche Diagnostics GmbH, Mannheim, Germany) on the automated analyzer Cobas 8000 (Roche). Levels of HbA1C were measured by ion exchange chromatography using the Arkray Adams HA-8180V analyzer (Arkray Corporation, Kyoto, Japan). C-peptide levels were determined with a commercially available kit (Immunotech, Marseille, France) using an immunoradiometric assay with specific antibodies.

Adiponectin was determined with the immunochemistry kit Human Adiponectin ELISA (Biovendor Laboratory Medicine Inc., Brno, Czech Republic) according to the manufacturer’s instructions. The antibodies used in this kit are specific for human adiponectin. The assay sensitivity was 26 ng/mL; the precision coefficients of variation (CVs) were 4.9% (intra-assay) and 6.7% (inter-assay).

Levels of A-FABP were assessed using the Human Adipocyte FABP4 ELISA kit (Biovendor Laboratory Medicine Inc.) according to the manufacturer’s instructions. The antibodies used in this ELISA are specific for human A-FABP. The assay sensitivity was 0.08 ng/mL; the precision CVs were 2.5% (intra-assay) and 3.9% (inter-assay).

Levels of FGF-19 were obtained with Human FGF-19 ELISA (Biovendor Laboratory Medicine Inc.) according to the manufacturer’s instructions. The antibodies used in this ELISA are specific for human FGF-19. The assay sensitivity was 4.8 pg/mL; the precision CVs were 6% (intra-assay) and 7.5% (inter-assay).

### 2.3. Statistical Analyses

All values were expressed as medians and interquartile ranges. Differences in variables between the groups were analyzed with the Mann–Whitney U-test. Differences between samples in individual trimesters were analyzed using the Wilcoxon signed-rank test. Analysis of covariates (ANCOVA) served to eliminate group differences in confounding factors. Spearman correlation analyses tested correlations between parameters in all groups. The Spearman coefficient (ρ) was used to express the value of correlation. Multivariate linear regression analyses were used for testing for independent associations between dependent and independent variables. Probability values of *p* < 0.05 were considered statistically significant. Statistica 14.0 software was used for statistical analyses.

## 3. Results

### 3.1. Baseline Clinical and Laboratory Parameters and Adipokine Levels in Individual Groups

A total of 23 pregnant women with early diagnosed GDM, 29 pregnant women without GDM, and 25 non-pregnant healthy controls met the eligibility criteria for this study. In the first trimester, women with GDM had a significantly higher weight, BMI, waist circumference, FPG, HbA1C, and C-peptide levels compared to both pregnant women without GDM and non-pregnant healthy controls. Pregnant women without GDM had significantly higher waist circumference and C-peptide levels, but lower FPG and HbA1C levels compared to non-pregnant healthy controls.

Levels of adiponectin were significantly decreased and levels of A-FABP were significantly elevated in women with GDM compared to both pregnant women without GDM and non-pregnant healthy controls. In contrast, A-FABP levels were decreased in women without GDM compared to non-pregnant healthy controls. There were no significant differences in FGF-19 between the groups. See Table 1. ANCOVA analysis showed the same significant differences in adipokines between groups even after adjusting for BMI.

### 3.2. Changes in Selected Clinical Parameters, Glucose Control, and Adipokines during Pregnancy

Changes in selected clinical parameters, glucose control, and adipokines during pregnancy are shown in Table 2. Lifestyle and behavioral changes were recommended to all patients with GDM during their first visit, with 30% of them (7 of 23) being treated with insulin (insulin therapy was initiated at a median of 15 {14–17} weeks). Because of the early therapy for GDM, there were no significant differences in weight, BMI, and HbA1C between women with and without GDM during the second and third trimesters compared to the first trimester. While women without GDM gradually increased their weight and BMI throughout their pregnancy, those with GDM showed a significant rise in their weight and BMI only in the third trimester. Those with GDM showed significantly higher waist circumference in the second trimester and higher fasting glucose levels only in the third trimester compared to women without GDM. As in the first trimester, no significant differences in SBP and DBP were found in the second and third trimesters.

Changes in adipokines reflected changes in body weight and BMI, see Table 2 and Figure 1. Adiponectin concentrations remained almost unchanged in the GDM group but dropped significantly in controls without GDM. As a result, there were no significant differences in adiponectin levels between the groups (GDM+ vs. GDM−) during the second and third trimesters. A-FABP decreased in women with GDM and increased in those without GDM during pregnancy. Thus, A-FABP was significantly higher in patients with GDM only in the second trimester while it did not differ between the groups in the third trimester (GDM+ vs. GDM−). No significant changes in FGF-19 were detected in both groups during pregnancy. FGF-19 levels did not significantly differ between the groups (GDM+ vs. GDM−) throughout the study.

### 3.3. Relationship of Adipokines to Clinical and Laboratory Parameters

Univariate correlation analyses of baseline data were performed in all pregnant women (*n* = 52). Adiponectin was significantly inversely (*p* < 0.05) correlated with body weight (ρ = −0.34), BMI (ρ = −0.39), waist circumference (ρ = −0.31), FPG (ρ = −0.38), and HbA1C (ρ = −0.23). A-FABP was positively correlated with body weight (ρ = 0.55), BMI (ρ = 0.52), waist circumference (ρ = 0.53), SBP (ρ = 0.17), FPG (ρ = 0.46), and HbA1C (ρ = 0.42). FGF-19 was inversely correlated with body weight (ρ= −0.23), BMI (ρ = −0.20), SBP (ρ = −0.39), and DBP (ρ = −0.32). There was also a significant inverse correlation between adiponectin and A-FABP levels (ρ = −0.18). FGF-19 was correlated with neither adiponectin nor A-FABP.

Table 3 shows the results of a multivariate regression analysis of independent factors affecting adiponectin, A-FABP, and FGF-19 as dependent variables based on data from all three trimesters in all pregnant women (*n* = 52). Adiponectin was independently associated with waist circumference and HbA1C; A-FABP with weight, SBP, and fasting glucose; and FGF-19 with SBP only.

## 4. Discussion

The present study demonstrated that women with early diagnosed GDM had significantly decreased concentrations of adiponectin and increased concentrations of A-FABP compared to pregnant women without GDM or non-pregnant healthy controls. Both adipokines correlated with visceral adiposity and glucose control. A-FABP was inversely correlated with adiponectin and both adipokines were associated with markers of visceral adiposity and glucose control. Multivariate regression analysis showed independent associations between adiponectin and waist and HbA1C, as well as between A-FABP and weight, SBP, and fasting glucose levels. Importantly, the introduction of early therapeutic intervention was associated with the shifting of circulating concentrations of these adipokines to levels comparable with those in healthy pregnant women.

In previous studies, adiponectin levels were found to be lower in women with GDM; additionally, hypoadiponectinemia in early pregnancy could be considered a predictive marker for GDM development [14,15,16,17,18]. A meta-analysis comprising 2865 pregnant women showed that pre-pregnancy and early pregnancy measurements of circulating adiponectin may improve the identification of women at high risk for developing GDM [19]. First trimester adiponectin concentrations had a pooled sensitivity of 60.3%, a specificity of 81.3%, and a diagnostic odds ratio of 6.6 for GDM prediction. However, there are also some conflicting findings. For example, Ebert et al. reported that adiponectin concentrations were lower in pregnant women compared to non-pregnant women but were not affected by GDM presence [20]. Discrepancies between studies may be due to different diagnostic criteria, population characteristics, pregnancy rates, and the application of treatment measures. Decreased adiponectin expression during pregnancy is thought to increase insulin resistance, leading to decreased glucose uptake. In the case of GDM, pancreatic cell dysfunction is unable to overcome insulin resistance, resulting in hyperglycemia [8]. The independent association between adiponectin and HbA1C in our study highlights the possible role of adiponectin in glucose control. The present study also shows that early implementation of lifestyle interventions not only attenuated weight gain but also prevented further decreases in adiponectin levels during pregnancy in women with GDM, shifting their adiponectin concentrations in the third trimester to levels similar to those seen in healthy pregnant women.

Several studies have found significantly elevated A-FABP in women with GDM [21,22,23,24,25,26,27,28,29], with some authors finding that elevated circulating A-FABP levels during the first trimester were associated with a higher risk of developing GDM [27,28]. Zhang et al. reported a trend toward increasing A-FABP levels in the second to the third trimester in patients with GDM [23]. We found this tendency in healthy pregnant women as well. A-FABP is secreted by adipocytes and is also released from the placenta in pregnant women [30]. Circulating levels of A-FABP are associated with lipolysis and rise due to insulin resistance during pregnancy. A-FABP overexpression in the placenta and decidua in GDM is stimulated by the action of placental lactogen, progesterone, and the synergistic effect of estrogen and progesterone whose levels are steadily elevated until delivery [29]. Early introduction of lifestyle interventions and appropriate GDM treatment was associated with a reduction in A-FABP in our study. Therefore, there were no significant differences between healthy pregnant women and patients with GDM in the third trimester. The independent associations between A-FABP and weight, fasting glucose, and SBP once again point to the role of insulin resistance in the development of GDM and its complications. A-FABP contributes not only to impaired glucose control but also to the development of gestational hypertension and preeclampsia [31]. Therefore, early GDM diagnosis and treatment are also important in preventing these complications.

Unlike some previous studies, we found no significant differences in FGF-19 between GDM patients and healthy pregnant controls throughout all trimesters. Wang et al. reported decreased circulating levels of FGF-19 in women with GDM diagnosed at 24–28 weeks of gestation [32]. There was also reduced placental and muscular expression of FGF-19 in women with GDM after delivery [33]. In contrast, a recent study found plasma concentrations of FGF-19 in the umbilical cord similar to those in healthy pregnant controls, suggesting that GDM does not affect fetal FGF-19 levels [34]. Wang et al. also demonstrated an independent and inverse association between FGF-19 and insulin resistance in GDM [32]. In our study, FGF-19 was inversely correlated with body weight, BMI, SBP, and DBP. In multiple regression analysis, however, it was independently associated with only SBP. We hypothesize that lifestyle interventions introduced to women with early-diagnosed GDM to avoid weight gain and increased insulin resistance could have prevented significant reductions in FGF-19 levels in the second and third trimesters. This may explain why there were no differences between GDM women and healthy controls during pregnancy. This finding supports the correlations of FGF-19 with body weight and BMI. The independent association between FGF-19 and SBP may thus suggest a role of FGF-19 in gestational hypertension, but this potential relationship needs to be elucidated in further studies.

One limitation of this study is the focus on a group of women with early diagnosed GDM only. Of course, there are women with normal glucose levels during the first trimester and diagnosed with GDM following an OGTT at 24–28 weeks of pregnancy; these make up the majority of GDM cases. Since varied manifestations of GDM may stem from different pathophysiological mechanisms (insulin resistance versus impaired insulin secretion), adipokine production may also differ for various GDM phenotypes. All women with GDM in our study were intervened early which could have affected their natural course of changes in adipokine levels (i.e., without intervention). Moreover, about a third of women in the GDM group were treated with insulin, potentially modifying the regulation of the above protein production. Finally, the study did not include a control group of women with GDM who had not undergone therapeutic intervention, therefore conclusion on the effect of the intervention is rather limited. The strength of this study lies in its prospective nature and longitudinal monitoring of adipokines levels, including the potential impact of early therapeutic interventions.

## 5. Conclusions

Women with GDM showed altered adipokine production already in the first trimester of pregnancy. They had increased A-FABP and decreased adiponectin levels correlating with visceral adiposity and glucose control. Early diagnosis of GDM necessitating the introduction of lifestyle interventions and early treatment was associated not only with the prevention of weight gain but also normalization of adipokines to levels similar to those in healthy pregnant controls. These findings support the importance of GDM screening in the early stages of pregnancy and the possible role of endocrine dysfunction of adipose tissue in the development of gestational diabetes.

## Figures and Tables

**Figure 1 jcm-11-02417-f001:**
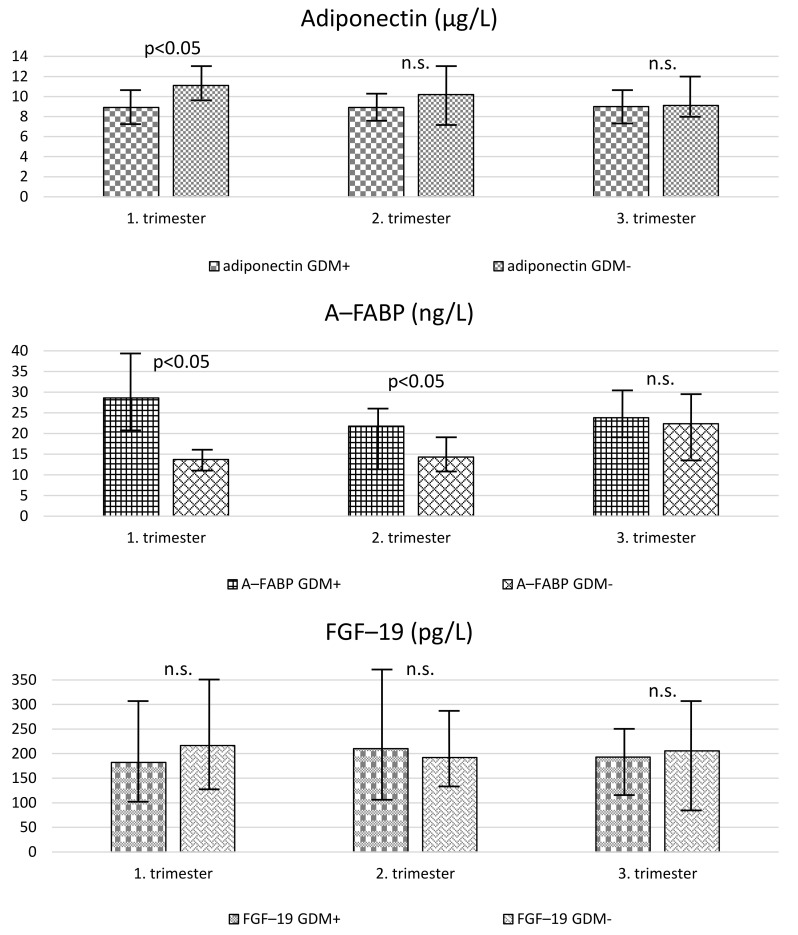
Levels of adipokines in different groups. GDM = gestational diabetes mellitus; A-FABP = adipocyte fatty acid-binding protein; FGF-19 = fibroblast growth factor-19. Values are expressed as medians (boxes) and interquartile ranges (whiskers).

**Table 1 jcm-11-02417-t001:** Baseline clinical and laboratory parameters in individual groups.

	Pregnant Women with GDM (*n* = 23)	Pregnant Women without GDM (*n* = 29)	Non-Pregnant Healthy Controls (*n* = 25)
age (years)	32.0 (27.1–36.2)	30.0 (27.9–32.3)	29.4 (26.6–33.0)
gestational age (weeks)	11.1 (10.2–11.8)	10.9 (9.9–11.6)	-
weight (kg)	80.2 (72.3–88.9) ^b,c^	65.1 (61.3–74.9) ^a^	63.6 (59.9–67.6) ^a^
BMI (kg/m^2^)	28.5 (25.4–32.8) ^b,c^	23.4 (20.4–25.1) ^a^	21.9 (20.3–23.9) ^a^
waist (cm)	96.3 (91.4–97.8) ^b,c^	80.5 (74.4–87.5) ^a,c^	75.5 (70.4–81.6) ^a,b^
SBP (mmHg)	121.9 (109.1–129.4)	120.2 (111.5–127.4)	117.1 (110.0–126.9)
DBP (mmHg)	76.3 (67.2–81.6)	78.0 (69.8–81.3)	74.2 (69.0–80.6)
FPG (mmol/l)	5.1 (4.9–5.3) ^b,c^	4.3 (4.1–4.4) ^a,c^	4.7 (4.5–4.8) ^a,b^
HbA_1C_ (mmol/mol)	32.3 (30.6–34.9) ^b,c^	30.0 (27.5–32.2) ^a,c^	31.0 (30.9–33.2) ^a,b^
C-peptide (pmol/l)	922.4 (533.5–1468.0) ^b,c^	602.1 (416.5–748.3) ^a^	510.5 (405.0–595.6) ^a^
adiponectin (µg/mL)	8.9 (7.2–11.1) ^b,c^	11.1 (9.8–13.4) ^a^	10.6 (9.3–12.6) ^a^
A-FABP (ng/mL)	28.6 (20.4–39.4) ^b,c^	13.7 (10.8–16.2) ^a,c^	20.3 (17.7–26.8) ^a,b^
FGF-19 (pg/mL)	182.2 (101.3–309.3)	216.4 (131.7–350.5)	266.6 (164.4–320.3)

GDM = gestational diabetes mellitus; BMI = body mass index; SBP = systolic blood pressure; DBP = diastolic blood pressure; FPG = fasting plasma glucose; HbA1C = glycated hemoglobin A1C; A-FABP = adipocyte fatty acid-binding protein; FGF-19 = fibroblast growth factor-19. Values are expressed as the median (25 and 75 percentile). Significant differences (*p* < 0.05) according to the Mann–Whitney U-test: ^a^ = vs. women with GDM; ^b^ = vs. women without GDM; ^c^ = vs. healthy controls.

**Table 2 jcm-11-02417-t002:** Changes in selected clinical parameters, glucose control, and adipokines during pregnancy in women with and without GDM.

		First Trimester (8–12 Weeks)	Second Trimester (24–28 Weeks)	Third Trimester (34–38 Weeks)
weight (kg)	GDM+	80.2 (72.3–88.9)	77.5 (68.0–93.9) ^c^	82.0 (77.2–90.0) ^b^
	GDM−	65.1 (61.3–74.9) ^b,c^ ¶	77.0 (71.2–82.2) ^a,c^	83.2 (74.7–90.5) ^a,b^
BMI (kg/m^2^)	GDM+	28.5 (25.4–32.8)	27.5 (23.7–32.4) ^c^	27.4 (25.8–33.8) ^b^
	GDM−	23.4 (20.4–25.1) ^b,c^ ¶	25.9 (23.6–28.0) ^a,c^	28.5 (25.1–31.0) ^a,b^
waist (cm)	GDM+	96.3 (91.4–97.8) ^c^	104.5 (98.3–110.1) ^c^	108.3 (103.2–114.4) ^a,b^
	GDM−	80.5 (74.4–87.5) ^b,c^ ¶	96.0 (90.2–102.5) ^a,c^ ¶	106.1 (98.7–114.5) ^a,b^
SBP (mm Hg)	GDM+	121.9 (109.1–129.4)	120.0 (112.5–129.6)	123.8 (117.3–132.1)
	GDM−	120.2 (111.5–127.4)	118.1 (108.9–120.4)	121.2 (114.0–130.5)
DBP (mm Hg)	GDM+	76.3 (67.2–81.6)	73.2 (69.5–76.5) ^c^	82.1 (74.7–85.3) ^b^
	GDM−	78.0 (69.8–81.3)	71.2 (64.9–75.1)	77.5 (88.0–81.0)
FPG (mmol/L)	GDM+	5.1 (4.9–5.3) ^b^	4.5 (4.4–5.0) a	4.8 (4.3–5.3)
	GDM−	4.3 (4.1–4.4) ¶	4.2 (4.0–4.9)	4.4 (4.1–4.5) ¶
HbA_1C_ (mmol/mol)	GDM+	32.3 (30.6–34.9) ^c^	31.0 (30.2–33.1) ^c^	36.0 (32.0–38.5) ^a,b^
	GDM−	30.0 (27.5–32.2) ^c^ ¶	29.5 (28.1–31.0) ^c^	33.3 (31.0–34.2) ^a,b^
adiponectin (µg/mL)	GDM+	8.9 (7.2–11.1)	8.9 (7.7–10.3)	9.0 (7.3–10.7)
	GDM−	11.1 (9.8–13.4) ^c^ ¶	10.2 (7.2–13.5)	9.1 (8.0–12.1) ^a^
A-FABP (ng/mL)	GDM+	28.6 (20.4–39.4) ^b,c^	21.8 (12.8–26.2) ^a^	23.8 (19.4–30.1) ^a^
	GDM−	13.7 (10.8–16.2) ^c^ ¶	14.3 (11.0–18.0) ^c^ ¶	22.4 (13.4–29.3) ^a,b^
FGF-19 (pg/mL)	GDM+	182.2 (101.3–309.3)	210.1 (104.6–360.0)	193.1 (105.1–250.4)
	GDM−	216.4 (131.7–350.5)	192.0 (139.6–289.3)	205.5 (87.4–302.0)

GDM = gestational diabetes mellitus; BMI = body mass index; SBP = systolic blood pressure; DBP = diastolic blood pressure; FPG = fasting plasma glucose; HbA1C = glycated hemoglobin A1C; A-FABP = adipocyte fatty acid-binding protein; FGF-19 = fibroblast growth factor-19. Values are expressed as the median (25 and 75 percentile). Significant differences (*p* < 0.05) according to the Wilcoxon signed-rank test: ^a^ = vs. 1st trimester; ^b^ = vs. 2nd trimester; ^c^ = vs. 3rd trimester or significant differences according to the Mann–Whitney U-test: ¶ = women with GDM (*n* = 23) vs. women without GDM (*n* = 29).

**Table 3 jcm-11-02417-t003:** Multivariate linear regression analysis of independent factors affecting adiponectin, adipocyte fatty acid-binding protein, and fibroblast growth factor-19 as dependent variables.

	**Adiponectin**					
	*Unstandardized coefficients*		*Standardized coefficients*		** *t* **	** *Sig.* **
	** *B* **	** *SE* **	** *Beta* **	** *SE* **		
weight	0.259	0.464	0.034	0.061	0.558	0.578
BMI	−0.621	0.418	−0.236	0.159	−1.486	0.140
waist	0.672	0.327	0.074	0.036	2.058	0.041
FPG	0.093	0.226	0.211	0.511	0.414	0.680
HbA_1C_	0.530	0.257	0.175	0.085	2.065	0.046
	**Adipocyte-fatty acid binding protein**					
	*Unstandardized coefficients*		*Standardized coefficients*		** *t* **	** *Sig.* **
	** *B* **	** *SE* **	** *Beta* **	** *SE* **		
weight	1.056	0.497	0.313	0.148	2.124	0.035
BMI	0.168	0.447	0.144	0.382	0.377	0.707
waist	−0.191	0.370	−0.047	0.091	−0.515	0.607
FPG	0.760	0.246	3.860	1.251	3.087	0.002
HbA_1C_	0.332	0.293	0.246	0.217	1.134	0.259
SBP	1.210	0.287	0.240	0.057	4.214	0.000
	**Fibroblast growth factor-19**					
	*Unstandardized coefficients*		*Standardized coefficients*		** *t* **	** *Sig.* **
	** *B* **	** *SE* **	** *Beta* **	** *SE* **		
weight	0.200	0.916	0.895	4.107	0.218	0.828
BMI	−0.242	0.879	−3.123	1.340	−0.275	0.783
SBP	0.722	0.753	2.157	2.250	0.959	0.039
DBP	−0.015	0.666	−0.072	3.146	−0.023	0.982

BMI = body mass index; SBP = systolic blood pressure; DBP = diastolic blood pressure; FPG = fasting plasma glucose; HbA1C = glycated hemoglobin A1C.

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
