# Peer review of "Adiponectin, A-FABP and FGF-19 Levels in Women with Early Diagnosed Gestational Diabetes"

_jcm, 2022, doi:10.3390/jcm11092417_

Round 1
Reviewer 1 Report
The idea of a search for markers of GDM is great, but the selection of control group makes the result of a lesser importance. There is a clear difference in the level of adipokines, FABP between subjects of different fat mass. So what is the point of checking sophisticated factors, when we clearly see that there is a difference in body mass?
Authors should consider selection of a different control group – comparable in weight/BMI – to exclude the clear confounders.
Therefore conclusions are overstatements.
Another lacking confounder seem to be thyroid function tests
Author Response
Thank you very much for your valuable comments.
Comment #1 The idea of a search for markers of GDM is great, but the selection of control group makes the result of a lesser importance. There is a clear difference in the level of adipokines, FABP between subjects of different fat mass. So what is the point of checking sophisticated factors, when we clearly see that there is a difference in body mass? Authors should consider selection of a different control group – comparable in weight/BMI – to exclude the clear confounders. Therefore conclusions are overstatements.
Response: You are right - there were significant differences in BMI of pregnant women with GDM compared to those without GDM. We believe that abdominal obesity (expressed by BMI and waist circumference) is one of the major risk factors (and maybe the crucial factor) for the development of early GDM, and therefore it is very difficult to find healthy pregnant controls (without GDM) of the same race and age with similar BMI values. However, we performed ANOVA statistical analysis to compare adipokines levels after adjusting for BMI. This analysis showed similar differences between groups even after adjusting for BMI. Now it is stated in the manuscript.
Comment #2 Another lacking confounder seem to be thyroid function tests.
Response: All study participants had normal thyroid function. Now it is stated in the manuscript.
The text of the manuscript was proofread of native speaker and was corrected.
Reviewer 2 Report
This descriptive study in women with early GDM (FPG 5.1-6.9 mmol/l) largely recapitulates data obtained in earlier studies.
I have one query:
- why was A-FABP lower in prgenant women without GDM vs. non-pregnant healthy controls? With A-FABP being expressed in placenta/decidua, one would expect the inverse.
Author Response
Thank you very much for your valuable comments.
This descriptive study in women with early GDM (FPG 5.1-6.9 mmol/l) largely recapitulates data obtained in earlier studies.
Comment #1 I have one query: - why was A-FABP lower in pregnant women without GDM vs. non-pregnant healthy controls? With A-FABP being expressed in placenta/decidua, one would expect the inverse.
Response: You are right - in the first trimester there were significant differences in A-FABP levels in pregnant women without GDM compared to non-pregnant healthy controls. It is likely that the proportion of placental/decidual secretion in A-FABP levels will be small at the beginning of pregnancy, while the presence of dysfunctional adipose tissue will be the main contributor to differences in its production. We think that the main reason why pregnant women without GDM have lower levels of A-FABP compared to non-pregnant healthy controls is in the stricter definition of GDM, i.e. in the definition for normal fasting glycaemia levels, which is associated with other metabolic parameters. GDM was defined as a fasting blood glucose level ≥5.1 mmol/l (so normal upper limit was 5.0 mmol/l), while in healthy non-pregnant women the normal upper limit is 5.5 mmol/l according to ADA/EASD criteria for diabetes or prediabetes. It is evident (see Table 1) that healthy non-pregnant women had also significantly higher levels of fasting glycaemia, glycated hemoglobin and (insignificantly) lower levels of adiponectin. The stricter criteria for the diagnosis of GDM distinguish as healthy pregnant women those with a more favorable metabolic profile (than non-diabetic non-pregnant controls).
The text of the manuscript was proofread of native speaker and was corrected.
Reviewer 3 Report
Manuscript: Adiponectin, A-FABP and FGF-19 levels in women with early
diagnosed gestational diabetes: the influence of therapeutic intervention
The authors have presented a quality manuscript, very well structured. The findings of the analysis are interesting and I believe that it lays the groundwork for future studies in this clinical area. I only have minor suggestions.
- Page 2, Lines 62-65: Please review sentence construction “Studies in animal models suggested that A-FABP is important in player in the regulation of glucose homeostasis, and deletion of the A-FABP gene protects mice from insulin resistance and hyperinsulinemia associated with obesity.”
- Methods: Please provide information on the study period. Specify when these patients were enrolled and How long did the follow-up last?
- Proofread manuscript, it requires some English editing. Some sentences are too long.
Author Response
Thank you very much for your valuable comments.
Comment # 1 : The authors have presented a quality manuscript, very well structured. The findings of the analysis are interesting and I believe that it lays the groundwork for future studies in this clinical area. I only have minor suggestions. Page 2, Lines 62-65: Please review sentence construction “Studies in animal models suggested that A-FABP is important in player in the regulation of glucose homeostasis, and deletion of the A-FABP gene protects mice from insulin resistance and hyperinsulinemia associated with obesity.”
Response: The sentence was corrected.
Comment # 2 Methods: Please provide information on the study period. Specify when these patients were enrolled and How long did the follow-up last?
Response: All pregnant women (both with or without GDM) were enrolled in the first trimester of pregnancy, followed for the rest of the pregnancy until delivery, when the glycemic disorder spontaneously normalized (that was a condition for the diagnosis of GDM). Now it is clearly stated in the manuscript.
Comment #3 Proofread manuscript, it requires some English editing. Some sentences are too long.
Response: The text of the manuscript was proofread of native speaker and was corrected.
Reviewer 4 Report
This work compares plasma levels of adiponectin, adipocyte fatty acid protein (A-FABP), and fibroblast growth factor 19 (FGF-19) in women with early diagnosed gestational diabetes, healthy pregnant women, and healthy non-pregnant women. The strength of the study is the longitudinal design - measurements of anthropometric parameters (body weight, BMI, waist circumference), metabolic parameters (glucose, glycated hemoglobin A1C) and adiponectin, A-FABP and FGF-19 levels were performed over all three trimesters of pregnancy.
The main problem of the article is the uncritical interpretation of the results. The authors should interpret the obtained results more critically throughout the article and especially in the discussion. For example, adiponectin levels in the first trimester of pregnancy were lower in women with gestational diabetes than in healthy pregnant and nonpregnant women - however, women with gestational diabetes also had a significantly higher BMI in the first trimester of pregnancy, and the authors did not rule out the possibility that increased BMI, rather than a diagnosis of GDM, was the cause of the decreased adiponectin. Furthermore, from the title to the discussion, the authors draw firm conclusions that therapeutic intervention in women with an early diagnosis of gestational diabetes prevents weight gain, improves glucose control, and normalizes adipokine levels. Because the study did not include a control group of women with gestational diabetes who had not undergone therapeutic intervention, a conclusion on the effect of the intervention should be formulated more carefully.
Another major problem is the statistical analysis used for the results shown in Table 3. The authors state that the test used is a multivariate logistic regression analysis, but the dependent variable is continuous rather than categorical.
Minor points:
1) The introduction is well structured, but I think the authors should cite more original scientific papers instead of scientific reviews (the introduction contains only one original scientific paper).
2) The laboratory analyzes are clearly described, but the authors should be more specific about the therapeutic intervention. What did the nutrition therapy, physical activity, and weight management include, and how was compliance with the instructions monitored?
3) Please specify the program used for statistical analysis.
4) In Table 1, the gestational age for pregnant women should be indicated.
5) The results on the concentration of adiponectin, A-FABP and FGF-19 in the first trimester of pregnancy are presented in three places - in Table 1, Table 3 and Figure 1. Also the concentrations of these molecules in the second and third trimesters of pregnancy are presented in Table 2 and Figure 1. In my opinion, it would be sufficient to present the results either in Table 2 or in Figure 1. If Figure 1 remains, please add statistical significance to it.
6) Lines 207-213: It is not clear on which data the respective analyzes were performed - whether on data from only one trimester of pregnancy or from all three trimesters and whether on subjects from all three groups or only on pregnant women (the number of subjects included in the analyzes should be specified).
7) References 4 and 5 need to be corrected (due to shifts after reference 4, all references in the text were shifted by +1).
8) Please add the number of women who received insulin therapy.
Author Response
Thank you very much for your valuable comments.
Comment #1 The main problem of the article is the uncritical interpretation of the results. The authors should interpret the obtained results more critically throughout the article and especially in the discussion. For example, adiponectin levels in the first trimester of pregnancy were lower in women with gestational diabetes than in healthy pregnant and nonpregnant women - however, women with gestational diabetes also had a significantly higher BMI in the first trimester of pregnancy, and the authors did not rule out the possibility that increased BMI, rather than a diagnosis of GDM, was the cause of the decreased adiponectin. Furthermore, from the title to the discussion, the authors draw firm conclusions that therapeutic intervention in women with an early diagnosis of gestational diabetes prevents weight gain, improves glucose control, and normalizes adipokine levels. Because the study did not include a control group of women with gestational diabetes who had not undergone therapeutic intervention, a conclusion on the effect of the intervention should be formulated more carefully.
Response: You are right - women with GDM had a significantly higher BMI compared to non-pregnant controls, or to women without GDM and BMI correlated both with adiponectin and A-FABP. Therefore, we performed ANOVA statistical analysis to compare adipokines levels after adjusting for BMI. And this analysis showed the same differences between groups even after adjusting for BMI. Now it is stated in the manuscript. Therapeutic intervention in women with an early diagnosis of GDM led to prevention of weight gain, and improved glucose control (this is the main reason why it is being introduced). And there were independent association between adiponectin and waist and HbA1c, and between A-FABP and weight, so we believe that this intervention also had an effect on adipokine levels. You are right – this study did not include a control group of women with GDM who had not undergone therapeutic intervention, which is one of its limitation (but this approach would not be acceptable from ethical point of view). This limitation is stated in the manuscript, now.
Comment #2: Another major problem is the statistical analysis used for the results shown in Table 3. The authors state that the test used is a multivariate logistic regression analysis, but the dependent variable is continuous rather than categorical.
Response: This is a typing error. The multiple regression analysis used was linear (not logistic), the appropriate manuscript correction was made.
Minor points:
Comment #3: The introduction is well structured, but I think the authors should cite more original scientific papers instead of scientific reviews (the introduction contains only one original scientific paper).
Response: You are right – however, we tried to meet the instructions for authors to use a limited number of references, thus we use, especially in the introduction, a summary of available information from review articles. On the contrary, in the discussion, where we discuss the individual details, the original papers dominate.
Comment #4: The laboratory analyzes are clearly described, but the authors should be more specific about the therapeutic intervention. What did the nutrition therapy, physical activity, and weight management include, and how was compliance with the instructions monitored?
Response: The concrete form of therapeutic intervention is now provided in the manuscript. The pregnant women were only asked how they follow the lifestyle change recommendations and re-educated as needed. No special compliance monitoring was performed.
Comment #5: Please specify the program used for statistical analysis.
Response: Statistica 14.0 software was used for statistical analyses.
Comment #6: In Table 1, the gestational age for pregnant women should be indicated.
Response: The gestational age for pregnant women was added in table 1.
Comment #7: The results on the concentration of adiponectin, A-FABP and FGF-19 in the first trimester of pregnancy are presented in three places - in Table 1, Table 3 and Figure 1. Also the concentrations of these molecules in the second and third trimesters of pregnancy are presented in Table 2 and Figure 1. In my opinion, it would be sufficient to present the results either in Table 2 or in Figure 1. If Figure 1 remains, please add statistical significance to it.
Response: We think that the data in Figure 1 provide a clearer overview of changes during pregnancy, the statistical significance was added.
Comment #8: Lines 207-213: It is not clear on which data the respective analyzes were performed - whether on data from only one trimester of pregnancy or from all three trimesters and whether on subjects from all three groups or only on pregnant women (the number of subjects included in the analyzes should be specified).
Response: Both analyses were performed – the comparison of variables between GDM+ vs GDM – in all trimesters and comparison of variables within the same group (GDM+ or GDM-) during pregnancy – see table 2. This may be better explained in the manuscript. The number of subjects was added.
Comment #9: References 4 and 5 need to be corrected (due to shifts after reference 4, all references in the text were shifted by +1).
Response: References were corrected.
Comment #10: Please add the number of women who received insulin therapy.
Response: The number (7 of 23) was added.
The text of the manuscript was proofread of native speaker and was corrected.
Round 2
Reviewer 1 Report
In my opinion, as stated in yhe firdt review, comparing groups that have different background that significantly affect results precludes attaing confident conclusions.
Matched group should be selected to obtain important conclusions
Author Response
Reviewer #1:
Thank you very much for your comments.
Comment #1 In my opinion, as stated in yhe firdt review, comparing groups that have different background that significantly affect results precludes attaing confident conclusions. Matched group should be selected to obtain important conclusions
Response: We enrolled age- and race-matched groups of women. We are not able to include other participants within 5 days (time determined by the editors for revision). However, we compared adipokines levels between groups also after BMI adjustment and the results were the same. Higher BMI in women with GDM compared to women without GDM was found in most of the original articles. For example, significant differences in BMI between GDM women and healthy pregnant controls have been documented in the following articles discussed in the manuscript:
Ortega-Senovilla H, Schaefer-Graf U, Meitzner K, Abou-Dakn M, Graf K, Kintscher U, Herrera E. Gestational diabetes mellitus causes changes in the concentrations of adipocyte fatty acid-binding protein and other adipocytokines in cord blood. Diabetes Care. 2011 Sep;34(9):2061-6. doi: 10.2337/dc11-0715. Epub 2011 Jul 20. PMID: 21775757; PMCID: PMC3161255.
Zhang Y, Zhang HH, Lu JH, Zheng SY, Long T, Li YT, Wu WZ, Wang F. Changes in serum adipocyte fatty acid-binding protein in women with gestational diabetes mellitus and normal pregnant women during mid- and late pregnancy. J Diabetes Investig. 2016 Sep;7(5):797-804. doi: 10.1111/jdi.12484. Epub 2016 Feb 24. PMID: 27181269; PMCID: PMC5009145.
Patro-Małysza J, Trojnar M, Kimber-Trojnar Ż, Mierzyński R, Bartosiewicz J, Oleszczuk J, Leszczyńska-Gorzelak B. FABP4 in Gestational Diabetes-Association between Mothers and Offspring. J Clin Med. 2019 Feb 27;8(3):285. doi: 10.3390/jcm8030285. PMID: 30818771; PMCID: PMC6462903.
Kimber-Trojnar Å», Patro-MaÅ‚ysza J, Trojnar M, SkórzyÅ„ska-Dziduszko KE, Bartosiewicz J, Oleszczuk J, LeszczyÅ„ska-Gorzelak B. Fatty Acid-Binding Protein 4-An "Inauspicious" Adipokine-In Serum and Urine of Post-Partum Women with Excessive Gestational Weight Gain and Gestational Diabetes Mellitus. J Clin Med. 2018 Dec 2;7(12):505. doi: 10.3390/jcm7120505. PMID: 30513800; PMCID: PMC6306707.
Wang D, Zhu W, Li J, An C, Wang Z. Serum concentrations of fibroblast growth factors 19 and 21 in women with gestational diabetes mellitus: association with insulin resistance, adiponectin, and polycystic ovary syndrome history. PLoS One. 2013 Nov 19;8(11):e81190. doi: 10.1371/journal.pone.0081190. PMID: 24260557; PMCID: PMC3834317.
Wang D, Xu S, Ding W, Zhu C, Deng S, Qiu X, Wang Z. Decreased placental and muscular expression of the fibroblast growth factor 19 in gestational diabetes mellitus. J Diabetes Investig. 2019 Jan;10(1):171-181. doi: 10.1111/jdi.12859. Epub 2018 Jun 6. PMID: 29734515; PMCID: PMC6319613.
Yang MN, Huang R, Liu X, Xu YJ, Wang WJ, He H, Zhang GH, Zheng T, Fang F, Fan JG, Li F, Zhang J, Li J, Ouyang F, Luo ZC. Fibroblast Growth Factor 19 in Gestational Diabetes Mellitus and Fetal Growth. Front Endocrinol (Lausanne). 2022 Jan 25; 12:805722. doi: 10.3389/fendo.2021.805722. PMID: 35145481; PMCID: PMC8821646.
Reviewer 4 Report
The authors were generally responsive and improved the manuscript.
However, I still have concerns.
The first relates to the title: since the study did not include a control group of women with gestational diabetes who did not undergo the intervention (for understandable ethical reasons), I think the part "the influence of the therapeutic intervention" should be removed from the title.
My other concerns relate to the statistical analyses.
The authors have corrected the Statistical Analyses and Results sections for typing error in the name of the multiple regression analysis used. However, they have now added a new confusion: in their response and in the text (line 153, lines 174-176) they state that they used "ANOVA post-hoc comparison" / "ANOVA analysis" to adjust for the influence of confounding factor BMI. ANOVA is used to test the difference between group means and is not a test that can adjust for the influence of confounding factors (there are other tests for that purpose).
Furthermore, the authors did not answer my question about the correlation analyzes described in lines 227-234 (lines 207-213 in an earlier version of the paper). Instead, they commented on the analyzes in Table 2, which are clearly described and were not the subject of my question.
For the correlation analyzes in lines 227-234, it is still unclear on which data they were performed. This is also not clear for the results in Table 3. Table 3 should include also summary data for each model (F, df, R, p) and indicate whether standardized or unstandardized beta values are shown. It is strange that some significant bivariate correlations are not consistent with the multiple regression results - for example, a parameter that is inversely correlated with a dependent variable (e.g., waist circumference with adiponectin level, r = -0.39) has a positive beta value in the multiple regression analysis (B=0.672).
Overall, I would recommend that the authors consult a statistician to verify the accuracy of their statistical analyzes.
One minor issue is the presentation of the same results in Table 2 and Figure 1, but that is a matter of the JCM journal policy.
Author Response
Reviewer #4:
Thank you very much for your comments.
Comment #1: The first relates to the title: since the study did not include a control group of women with gestational diabetes who did not undergo the intervention (for understandable ethical reasons), I think the part "the influence of the therapeutic intervention" should be removed from the title.
Response: The requested change has been made.
Comment #2: The authors have corrected the Statistical Analyses and Results sections for typing error in the name of the multiple regression analysis used. However, they have now added a new confusion: in their response and in the text (line 153, lines 174-176) they state that they used "ANOVA post-hoc comparison" / "ANOVA analysis" to adjust for the influence of confounding factor BMI. ANOVA is used to test the difference between group means and is not a test that can adjust for the influence of confounding factors (there are other tests for that purpose).
Response: You are right. It was not ANOVA, but analysis of covariates (ANCOVA) served to eliminate group differences in BMI. Now, it has been corrected in the manuscript.
Comment #3: Furthermore, the authors did not answer my question about the correlation analyzes described in lines 227-234 (lines 207-213 in an earlier version of the paper). Instead, they commented on the analyzes in Table 2, which are clearly described and were not the subject of my question. For the correlation analyzes in lines 227-234, it is still unclear on which data they were performed. This is also not clear for the results in Table 3.
Response: I'm sorry, but the lines were moved due to language correction, thus other results were commented. Univariate correlation analyses of baseline data (i.e. first trimester values) in all pregnant women (n=52, both GDM+ (n=23) and GDM- (n=29)) were performed. The multivariate linear regression analyses were performed in all pregnant women, based on data from all three trimesters. It is stated now in the manuscript.
Comment #4: Table 3 should include also summary data for each model (F, df, R, p) and indicate whether standardized or unstandardized beta values are shown. It is strange that some significant bivariate correlations are not consistent with the multiple regression results - for example, a parameter that is inversely correlated with a dependent variable (e.g., waist circumference with adiponectin level, r = -0.39) has a positive beta value in the multiple regression analysis (B=0.672).
Response: The table 3 has been corrected and both coefficients and SE have been added, now. You are right, in some cases several predictors have the opposite sign in multiple regression analysis compared to univariate correlation analysis. However, these are statistically insignificant predictors, according to our statistician a huge contribution of significant predictors can cause this situation in the regression equation.